# Production, purification, characterization, and biological properties of Rhodosporidium paludigenum polysaccharide

**Mengjian Liu** [ORCID]<sup></sup>, **WenJu Zhang**\*<sup></sup>, **Jun Yao**<sup></sup>, **Junli Niu**<sup></sup>

College of Animal Science and Technology, Shihezi University, Shihezi, Xinjiang, China

☯ These authors contributed equally to this work.
* zhangwj1022@sina.com

**Data Availability Statement:** All relevant data are within the paper.

**Funding:** National Key R&D Program of China (2018YFD0502100) https://service.most.gov.cn/

## Abstract

The yield of marine red yeast polysaccharide (MRYP) obtained from Rhodosporidium paludigenum was increased by optimizing fermentation conditions, and the pure polysaccharide was extracted by column chromatography. The molecular weight of pure MRYP and the ratio of mannose to glucose in components of MRYP were determined. Antioxidant and antibacterial abilities of MRYP were investigated *in vitro* and *in vivo*. The optimal fermentation parameters were as follows: Medium 4, pH = 6.72, temperature = 30.18˚C, blades speed = 461.36 r/min; the optimized yield reached 4323.90 mg/L, which was 1.31 times the original yield. The sequence of factors that affected the MRYP yield was the blades speed>pH>temperature. The main components of MRYP were MYH-1 and MYH-2. The molecular weights of MYH-1 and MYH-2 were 246.92 kDa and 21.88 kDa, respectively; they accounted for 53.60% and 28.75% of total polysaccharide. In MYH-1 and MYH-2, the proportion of glucose and mannose accounted for 46.94%, 38.46%, and 67.10%, 7.17%, respectively. *In vitro*, the ability of scavenging DPPH•, •OH, and •$O_2^-$ radical was 32.26%, 24.34%, and 22.09%; the minimum inhibitory concentration (MIC) of MRYP was 480 μg/mg. *In vivo*, MRYP improved the lambs' body weight, antioxidant enzyme activity, and the number of probiotics, but it reduced the feed/gain (F/G) ratio and the number of pathogenic bacteria in 60-days-old lambs.

## Introduction

Marine red yeast Rhodosporidium paludigenum is a kind of single-celled aquatic organism, which is widely distributed in the sea and various rivers. Recent studies have shown that marine microorganisms have superior characteristics compared to their terrestrial counterparts. First of all, marine yeasts showed higher tolerance to all the inhibitors tested than terrestrial yeasts, and many reports have revealed that most of marine yeasts strains showed excellent fermentation capability compared with the reference terrestrial yeast and other different yeasts species [1]. Besides, marine microorganisms, including marine yeasts, live in extreme environments, and this provides a unique potential for the synthesis of functional

**Competing interests:** The authors have declared that no competing interests exist.

biomolecules. Recent research showed that marine yeasts have several unique and promising features over the terrestrial yeasts, for example higher osmosis tolerance, higher special chemical productivity and ability to produce industrial enzymes. These indicate that marine yeasts have a great potential to be applied in various industries [2]. Last but not least, it has been reported that marine yeasts are able to produce many bioactive substances, such as amino acids, glucans, glutathione, toxins, enzymes, phytase, and vitamins with potential application in the food, pharmaceutical, cosmetic, and chemical industries as well as for marine culture and environment protection [3].

Compared to the marine red yeast, a crude polysaccharide of Rhodosporidium paludigenum (MRYP) is more widely used as a probiotic for immune-enhancement and growth-promotion in local farms, such as pig farms, layer farms, and cattle farms, which are situated in Xinjiang, China. This implies that MRYP might be a prebiotic, which plays an important role in biological activities and has potential developmental prospects. Several studies have reported that yeast polysaccharides have various bioactive properties, including antibacterial, antioxidant, and anti-inflammatory activities [4]. The biological effect of MRYP might be as follows: 1. The yeast polysaccharide lowers attachment ability of pathogens in the intestine [5]; 2. Innate immunity of intestinal surface is regulated through pathogen-associated molecular patterns of yeast polysaccharide [6]; 3. The immune overreaction and inflammation caused by a lipopolysaccharide (LPS) in the intestinal region is inhibited and relieved by MRYP [7]. Besides, the antigen-presenting cells (APC) such as dendritic cells and macrophages are affected by the yeast polysaccharide [8]. Therefore, we deduce that MRYP is much better than the ocean red yeast.

In theory, the molecular weight is one of important indicators of bioactivity of yeast polysaccharide; the molecular weight of 50–200 kDa indicates a high biological activity [9]. Therefore, the bioactivity of MRYP should be evaluated by determining its molecular weight. Mannosan and $\beta$-glucan were the most important bioactive components in the yeast polysaccharide [10]. Glucan increased the phagocytic activity of biologically active compounds, such as TNF-$\alpha$, IL-1, and NO [11]. Mannosan inhibited colonization of pathogenic bacteria and also increased the number of probiotics in the intestine [12, 13]. To indicate the bioactivity of MRYP components, we determined the proportion of mannose to glucose in MRYP.

In the past years, several studies have been undertaken to determine the microbial characteristics, bacteriostatic ability, and immunogenicity of oceanic red yeast. To the best of our knowledge, no previous study has examined the fermentation, characterization, and biological function of MRYP. In this study, firstly, we used single factor tests and Box–Behnken design to optimize the fermentation conditions. Secondly, MRYP was extracted and purified by DEAE-52 cellulose and Sephadex G-100 column chromatography. The molecular weight of MYH-1 and MYH-2, which were the components of MRYP, were determined by liquid chromatography. The ratio of mannose and glucose was also determined by liquid chromatography (mannose and glucose were the components of MYH-1 and MYH-2). Finally, the antioxidant and antibacterial activities of MRYP were evaluated *in vitro* and *in vivo*. This work provides a theoretical foundation to a wide range of possible application of MRYP in the industrial production, design, and operation.

## Materials and methods

### Animals and strains

The test lambs: One hundred newborn Suffolk-Kazakh lambs (aged 32 days, 2:1 male: female ratio) were chosen as experimental animals in Asar farming cooperatives, Solbastow, Changji Hui Autonomous Prefecture, China. All experimental lambs in this research have been

approved by the Animal Welfare Committee of Shihezi University (Xinjiang, China) with the ethical code: A2019-156-01.

In the antioxidant enzyme activity test, three blood samples (10 mL each) were collected randomly from marginal ear vessels of 10 lambs in each group without anesthesia, euthanasia, or any kind of animal sacrifice in this part of the study.

The experimental strain of marine red yeast Rhodosporidium paludigenum was obtained from the Yarkand River, Maigaiti, Kashi, China. The strain was isolated with a YPD solid medium with addition of sugar-cane molasses and identified by multiple sequence alignment (26S rRNA D1/D2) and phylogeny analysis.

The test strains for the antibacterial activities experiment: The Gram-positive Staphylococcus aureus, Gram-negative Escherichia coli and Salmonella enteric bacteria strains were purchased from the College of Animal Science and Technology, Shihezi University, Shihezi, China.

**Optimization of fermentation medium.** The fundamental fermentation parameters were as follows: pH = 7, 32°C, 300 r/min, and aerobic incubation for 72 h [14]. Each of the fermentation media was used as a single factor to produce MRYP. An automatic fermentation system with a 12 L fermentor was used as the experimental equipment.

Six different fermentation media were used as follows: (1) The yeast peptone dextrose adenine (YPDA) medium: yeast extract 10 g, peptone 20 g, glucose 20 g, and distilled water 1 L [15]; (2) Czapek medium: sucrose 30 g, sodium nitrate 3 g, dipotassium phosphate 1 g, potassium chloride 0.5 g, magnesium sulfate 0.5 g, ferrous sulfate 0.01 g, and distilled water 1 L [16]; (3) Medium 1: sucrose 50 g, soybean peptone 40 g, yeast extract 30 g, glycerin 16 g, and distilled water 1 L [17]; (4) Medium 2: sucrose 50 g, glucose 20 g, yeast extract 50 g, ammonium sulfate 10 g, potassium dihydrogen sulfate 3 g, glycerol 2 g, calcium chloride 1 g, and distilled water 1 L [18]; (5) Medium 3: sucrose 60 g, yeast paste 40 g, ammonium sulfate 10 g, potassium dihydrogen sulfate 3 g, glycerin 10 g, and distilled water 1 L [19]; (6) Medium 4: sucrose 50 g, glucose 20 g, peptone 60 g, ammonium sulfate 30 g, glycerin 3 g, calcium chloride 3 g, magnesium sulfate 1 g, calcium D-pantothenate 0.6 g, and distilled water 1 L. All the media were sterilized at 121°C in an autoclave sterilizer for 30 min.

**Optimization of fermentation pH, temperature, blades speed.** pH was used as a single factor in this study. In the primary screening test, the medium pH was set to 3.0, 4.0, 5.0, 6.0, 7.0, and 8.0. In the second screening test, based on the primary pH scope, the medium pH was set to 5.0, 5.4, 5.8, 6.2, 6.6, and 7.0. All the processes were repeated three times.

**Optimization of fermentation temperature.** The temperature, which was used as a single factor, was set to 24, 26, 28, 30, 32, and 34°C. All the processes were repeated three times.

**Optimization of fermentation blades speed.** The blades speed was used as a single factor in this study and set to 200, 300, 400, 500, 600, and 700 r/min [20]. All the processes were repeated three times.

**Box-Behnken design of fermentation conditions.** Based on the single factor experiments, we developed a Box-Behnken design (BBD) of three factors and three levels. Our main goal was to further optimize fermentation conditions. Three independent variables, i.e. fermentation pH, temperature, and blades speed, were designated as $X_1$, $X_2$, and $X_3$, respectively [21]. As shown in Table 1, the ranges of values were based on the results of preliminary experiments. The response variables indicated the polysaccharides yield.

**Validation of the mathematical model.** Based on the optimum fermentation conditions, triplicate fermentation experiments were performed to verify accuracy and availability of the mathematical model representing MRYP production. Moreover, the relative error between the actual and theoretical yield was calculated [22].

**Table 1. Independent variables and levels in Box-Behnken design.**

| | A | B | C |
|---|---|---|---|
| Coding value | pH | Temperature/°C | Blades speed/(r/min) |
| -1 | 5.8 | 28 | 500 |
| 0 | 6.2 | 30 | 600 |
| 1 | 6.6 | 32 | 700 |

**Extraction of MRYP.** The fermented liquor was centrifuged in a high-speed refrigerated centrifuge with 6000 r/min for 10 min at 4°C, and the supernatant was removed [23]. Three volumes of cold (20°C) distilled water were added to dissolve precipitates within 30 min of agitation. Neutral protease (300 IU/g) was added into the slurry, and it was incubated at a constant temperature of 40°C for 36 h [24]. After centrifugation (6000 rpm, 10 min), the sediment was collected and treated with sodium hydroxide (2%) for 3 h, and then it was neutralized with hydrochloric acid (5%) [25]. Two volumes of ethyl alcohol were added to wash the sediment, and this process was repeated thrice. Finally, the crude polysaccharides were obtained after freeze-drying the sediment for 48 h and grinding it to a powder [26]. The crude polysaccharides were weighed accurately with an electronic balance in three repeated trials.

**Purification of MRYP.** Two volumes of acetone were added to the crude MRYP, which was extracted by the method above. The mixture was agitated vigorously for 4 hours, and then it was centrifuged at 6000 r/min for 10 min. The supernatant was removed. The sediment was collected, and then it was deproteinated with Savage reagent (1-butanol/chloroform, v/v = 1:4) at 4°C for 24 h. The sediment was collected [27], added into a dialysis bag (3500 Da, 0.22 μm) and mixed with 5 volumes of distilled water for 48 h [28]. Finally, the sediment was lyophilized at -80°C for 48 h, and the pure MRYP was obtained.

**Isolation of MRYP.** Pure MRYP (200 mg) was dissolved in 10 mL of distilled water, and 2 mL of MRYP solution was passed through an anion exchange column (2.6 cm×50 cm) with DEAE-52 cellulose. The equilibrium liquid was a borate solution (0.05 mol/L, pH = 9.18). The eluent was a gradient sodium chloride solution at concentrations of 0.1, 0.5, 0.9, 1.3, and 1.7 mol/L [29]. In the anion exchange column, the flow velocity was 1 mL/min. Fractions of pure MRYP were eluted with different concentrations of NaCl. The eluents were collected in 6 mL aliquots and desalted for further purification on the Sephadex G-100 column, with distilled water as the equilibrium liquid at 0.5 mL/min [30]. The major polysaccharide fractions were collected, concentrated, and lyophilized at -80°C for 48 h, and all fractions of pure MRYP were obtained [31].

**Determination of the molecular weight.** The molecular weight of MRYP fractions was determined by high-performance gel permeation chromatography (HPGPC) with an Agilent 1200 HPLC system, which was equipped with a TSKgel-G3000-PWXL (7.8 mm×300 mm) column. Standard dextrans (404 kDa, 212 kDa, 112 kDa, 47.3 kDa, 22.8 kDa, 11.8 kDa, 5.9 kDa, and 2.5 kDa) were used for obtaining the standard curve [32]. The polysaccharide sample solution, which had a concentration of 5 mg/mL, was prepared by mixing the pure polysaccharide with the mobile phase [33]. Furthermore, 20 μL of polysaccharide sample solution was injected into the sample system, and the operation conditions were as follows: sample size was 20 μL; mobile phase was 0.1 mol/L sodium sulfate; column temperature was 35°C; flow velocity was 0.5 mL/min; and the maximum pressure was 25 bar [34].

**Analysis of monosaccharides composition.** According to the analysis method reported by Xutingting [35], the content of mannose and glucose in MRYP components MYH-1 and MYH-2 was determined by high-performance liquid chromatography (HPLC). Firstly, pure mannose and glucose were weighed precisely and mixed with distilled water separately. The

concentration of the standard solution was diluted down to 200 μg/mL, 400 μg/mL, 600 μg/ mL, 800 μg/mL, and 1000 μg/mL. We injected 50 μL of each standard solution into the sample inlet, and the regression equation was obtained accordingly. Secondly, each sample of MRYP was weighed precisely and mixed with distilled water and injected into the sample inlet. The operation conditions of HPLC were as follows: the mobile phase was deionized water; flow velocity was 0.5 mL/min; column temperature was 85˚C; and the sample size was 50 μL [36].

**Antioxidant activity of MRYP.** *1) Scavenging of DPPH radical (DPPH•).* The polysaccharide solution was diluted down to 100 μg/mL, 200 μg/mL, 400 μg/mL, 600 μg/mL, 800 μg/mL, 1000 μg/mL, and 1200 μg/mL. Then 2 mL of polysaccharide solution and the control sample were separately mixed with 2 mL of ethanolic DPPH• (1,1-diphenyl-2-picrylhydrazyl) (5 mmol/L) [37]. The mixture was shaken violently and incubated in the dark at 25˚C for 30 min. The absorption of the samples was measured with an ultraviolet spectrophotometer at λ = 517 nm [38]. Each processing was repeated three times, and the average scavenging activity was calculated by the method explained in paragraphs (2) and (3).

The scavenging activity was calculated as follows:

$$Scavenging\,activity = [1 - (A0 - A1)/A2] \times 100\%$$

Where A0 is the absorbance of the mixture of DPPH solution and the sample; A1 is the absorbance of ethanolic DPPH• mixed with deionized water; and A2 is the absorbance of DPPH solution mixed with deionized water.

*2) Scavenging of hydroxyl radical (•OH).* The $FeSO_4$ solution (1.0 mL, 9.0 mmol/L) was mixed with $H_2O_2$ (1.0 mL, 0.06%), deionized water (5 mL), sodium salicylate (1.0 mL, 9.0 mmol/L), and 1.0 mL of sample polysaccharide solution of above concentrations (100–1200 μg/mL), and then the mixture was incubated at 37˚C for 1 h [39]. The absorption of the samples was measured with a microplate spectrophotometer at λ = 517 nm [40].

The antioxidant activity was calculated by using the following equation:

$$Scavenging\,activity = [1 - (A0 - A1)/A2] \times 100\%$$

Where A0 is the absorbance of the sample as mentioned above; A1 is the absorbance when the $H_2O_2$ was replaced by the same volume of deionized water; and A2 is the absorbance when the sample polysaccharide solution was replaced by the same volume of deionized water.

*3) Scavenging of superoxide anion free radical (•$O_2^-$).* The Tris-hydrochloric acid (3 mL, pH = 8.2, 25˚C) was mixed with pyrogallol (3 mL) and precisely incubated at 25˚C for 4 min. Then the incubation was terminated by mixing with hydrochloric acid (1 mL, 19 mmol/L). The absorption of the samples was measured on the microplate spectrophotometer at λ = 420 nm [41]. The scavenging activity was calculated from the following equation:

$$Scavenging\,activity = [1 - (A0 - A1)/A2] \times 100\%$$

Where A0 is the absorbance of the sample mentioned above; A1 is the absorbance when the sample polysaccharide solution was replaced by the same volume of deionized water; and A2 is the absorbance when the Tris-hydrochloric acid was replaced by the same volume of deionized water.

**Antibacterial activity of MRYP.** Each culture suspension (200 μL) of bacteria (Salmonella, Escherichia coli and Staphylococcus aureus) was inoculated onto the surface of Luria-Bertani agar medium. The polysaccharide solution was diluted down to 30 μg/mL, 60 μg/mL, 120 μg/mL, 240 μg/mL, 480 μg/mL, 1000 μg/mL, 1200 μg/mL, and the positive control sample of amoxicillin (200 μg/mL) was prepared [42]. Then 50 μL of diluted polysaccharide solution and positive control sample were loaded into holes (9 mm in diameter), which have been

punched in the agar layer. The agar plates were kept at 4˚C for 1 h and then incubated at 37˚C for 24 h [43]. The antimicrobial activity was evaluated by determining the zone of growth inhibition (diameter expressed in centimeter) around the holes and the minimum inhibitory concentration was obtained [44]."-" represents no antibacterial activity [45]. Each test was repeated three times.

**Effects of MRYP on lambs *in vivo*.**    The experiments were conducted in the Asar farming cooperatives (Changji, China). The experiments period was 28 days from 24[th] December 2019 to 20[th] January 2020.

One hundred lambs were used in a 28-day randomized, complete block design experiment. The experimental lambs were split into five groups, and there were twenty lambs in each group. Following a week of adaption, the lambs were fed artificially. The feed period was 28 days. At the beginning of the experiment, the lambs were 32 days old. At the end of the experiment, the lambs were 60 days old. The groups were designed as follows: control group (milk replacement), antibiotic group (milk replacement + 0.005% gentamicin), MRYP 0.1% group (milk replacement + 0.1% MRYP), MRYP 0.3% group (milk replacement + 0.3% MRYP), and MRYP 0.5% group (milk replacement + 0.5% MRYP). The only additive in lamb feed is the single variable in this experiment.

a. *Body weight test*. The total feed intake of every group was counted from the 17[th] to 60[th] day. At the end of the feeding experiment, each experimental lamb was weighed, and the feed conversion ratio (F/G) was calculated by using the following equation [46]:

$$F/G(\%) = [Average daily feed intake(Kg)/Average daily gain(Kg)] \times 100\%$$

b. *Antioxidant enzyme activity test*. At the end of the feeding experiment, three blood samples (10 mL each) were collected randomly from marginal ear vessels of 10 lambs in each group. For determining the antioxidant activity of GSH-Px, T-SOD, MAD, and T-AOC in the serum, we used an assay kit [47].

c. *Bacteria quantity test*. The feces were collected randomly in the morning of the 60[th] day. The number of bacteria in feces was counted by the plate counting method. The culture media were as follows: MRS for Bifidobacterium and Lactobacillus, VRBGA for Escherichia coli, LB for Salmonella and Staphylococcus aureus [48]. The Bifidobacterium was incubated in a hostile anaerobic environment of 37˚C for 72 h [49]. Other bacteria were incubated in an aerobic environment with the same culture conditions. An automated bacterial colony counter was used to count the number of bacterial colonies.

## Statistical analysis

In this study, all values were expressed as the mean ± standard deviations, and the differences between means were considered significant at p < 0.05 by SPSS 18.0 software version 18 (SPSS Inc., New York, NY, USA). The differences among groups were analyzed by Duncan's multiple range test [50]. Design-Expert Software (Version 8.0.6, State Ease Inc., Minneapolis, NE, USA) was used to analyze the variance (ANOVA) and the parameters of the response equation. The goodness of fit of the polynomial model was evaluated by the coefficient of determination $R_2$ and Fisher's F-test [51]. Surface and contour plots illustrated the fitted polynomial equation, which showed the optimal fermentation conditions and the relationship between independent levels.

**Table 2. Effect of five fermentation media on the MRYP yield (mg/L).**

| Culture medium | YPDA medium | Czapek medium | Self-1 matching medium | Self-2 matching medium | Self-3 matching medium | Self-4 matching medium |
|---|---|---|---|---|---|---|
| Yield of polysaccharide | 2511.42±30.21a | 2212.93±22.86a | 2880.04±20.01b | 2915.68±22.97b | 2927.07±19.75b | 3301.21±12.93c |

Values are means ± standard deviations. In the same row, values with different small letter superscripts mean significant difference ($P<0.05$), different capital letter superscripts mean significant difference ($P<0.01$). The same as below.

## Results

Here, we investigated the effects of the fermentation medium, pH, blades speed, and temperature on the MRYP yield. Then we determined the optimal fermentation conditions for MRYP in Box–Behnken design. Besides, the molecular weight and monosaccharides composition were analyzed by HPGPC and HPLC, respectively. At last, the antioxidant and antibacterial activity were evaluated in *vitro* and in *vivo*.

### Effect of medium on the yield of MRYP

Table 2 presents the yields of MRYP in each medium. Based on the preliminary single-factor experiments, the yield in Medium 4 was 3301.21 mg/L, which was significantly higher than others ($p<0.05$).

This result indicated that the ingredients of Medium 4 might have a promoting effect on the production of MRYP. Therefore, Medium 4 was chosen for subsequent experimental optimization.

### Effect of pH on the yield of MRYP

pH of the fermentation medium is a crucial factor in the production of yeast polysaccharides. This is because pH affects yeast metabolite production and physiology. Besides, potentials of both inner and outer membranes are affected by the pH of fermentation medium.

Table 3 shows the yields of MRYP at different pH values. In the primary screening test, the yield of MRYP increased with a rise in pH until it reached its peak value (3746.32±74.10 mg/L) at pH = 6.0, which was significantly greater than others ($p<0.05$). Then the yield of MRYP declined until the end, that is, until the pH value reached 8.0. As seen in Table 3, the yield of MRYP was significantly higher than others when the pH range was from 5 to 7 ($p<0.05$). Therefore, the pH range of 5–7 was chosen for the second screening test.

In the second screening test, the yield of MRYP was significantly higher than others ($p<0.05$) when the pH range was between 5.8 and 6.6, and the peak value was 4102.18±76.66 mg/L when the pH was 6.2. Therefore, pH = 5.8, 6.2, and 6.6 were chosen for the fermentation medium.

**Table 3. Effect of different pH on the yield of MRYP (mg/L).**

| Preliminary screening | 3.0 | 4.0 | 5.0 | 6.0 | 7.0 | 8.0 |
|---|---|---|---|---|---|---|
| Yield of polysaccharide | 1142.51±71.76a | 2723.73±61.18c | 3385.38±67.41e | 3746.32±74.10f | 3138.24±63.94d | 2488.03±75.66b |
| Secondary screening | 5 | 5.4 | 5.8 | 6.2 | 6.6 | 7 |
| Yield of polysaccharide | 3431.71±66.08c | 3618.82±58.66d | 3644.78±55.71c | 4102.18±76.66e | 3931.34±89.48d | 3073.46±65.31a |

**Table 4. Effect of different blades speed on the yield of MRYP (mg/L).**

| Blades speed (r/min) | 200 | 300 | 400 | 500 | 600 | 700 |
|---|---|---|---|---|---|---|
| Yield of polysaccharide | 1970.08±13.72a | 3061.33±49.30b | 3535.75±32.56d | 3775.36±63.08e | 3647.10±53.83e | 3291.62±56.83c |

## Effect of blades speed on the yield of MRYP

Table 4 shows the yield of MRYP with respect to the different blades speed. As shown in Table 4, both too high and too low values of blades speed led to a lower yield of MRYP. The reasons for this phenomenon are as follows: firstly, a low blades speed causes a low oxygen content in the fermentation liquid, which cannot afford enough oxygen for breeding and growing of yeast. Secondly, a high blades speed causes a huge relative speed between blades and the fermentation liquid, which destroys yeast cells because of the additional shear forces. As shown in Table 4, 300–500 r/min was the appropriate range of blades speed because the MRYP yield was significantly higher than others ($p<0.05$) in this range. The peak yield was 3775.36 mg/L when the blades speed was 500 r/min. Therefore, the blades speed of 400, 500, and 600 r/min were selected for the response surface experiment.

## Effect of temperature on the yield of MRYP

As shown in Table 5, the yield of MRYP increased with a rise in temperature until it reached the peak value (3585.21±87.36 mg/L) at 30°C, which was significantly higher than others ($p<0.05$). Then the yield of MRYP declined until the end, that is, until the temperature reached 34°C. Therefore, the temperature of 28, 30, and 32°C were selected for the response surface experiment.

## Optimization by response surface methodology

In this study, response surface methodology (RSM) was used for precise optimization of fermentation conditions: pH (A), blades speed (B), fermentation temperature (C). According to the results of single-factor experiments (section 2.1 to 2.4), three independent variables were set for three selected variables (−1, 0, +1). Table 6 shows the combinations of different variables and their MRYP yields. Table 7 shows that the experimental outcomes corresponded to every independent variable, and we calculated the coefficients of the regression equation. Finally, by using multivariate linear regression analysis, we developed a mathematical model to determine the optimum fermentation conditions.

**Mathematical model building and significance testing.** Based on the preliminary single-factor experiments, the following conditions were chosen for the RSM experiments: fermentation pH (A) 5.8, 6.2, 6.6; blades speed (B) 400, 500, 600 r/min, and fermentation temperature (C) 28, 30, and 32°C. As shown in Table 6, when the fermentation pH was 6.2, temperature = 30°C, and blades speed = 500 r/min, the yield of MRYP reached its peak value of 4440.81±141.02 mg/L, which was significantly higher than others ($p<0.05$).

Multiple regression analysis was used to develop a mathematical model that determined the optimum fermentation conditions and to study the relationship between the response variable

**Table 5. Effect of different temperature on the yield of MRYP (mg/L).**

| Temperature(°C) | 24 | 26 | 28 | 30 | 32 | 34 |
|---|---|---|---|---|---|---|
| Yield of polysaccharide | 2162.83±86.68a | 3052.78±59.65c | 3118.66±66.87c | 3585.21±87.36e | 3423.06±15.00d | 2736.28±104.34b |

**Table 6. Response surface test of fermentation conditions.**

| No. | A | B | C | |
|---|---|---|---|---|
| | pH | Temperature (˚C) | Blades speed (r/min) | Yield of polysaccharide (mg/L) |
| 1 | 5.8 | 32 | 500 | 3031.54±102.03abcd |
| 2 | 5.8 | 28 | 500 | 3281.38±146.31cd |
| 3 | 6.6 | 32 | 500 | 2909.65±185.02abc |
| 4 | 6.6 | 28 | 500 | 3162.01±153.66bcd |
| 5 | 5.8 | 30 | 600 | 4159.37±110.74e |
| 6 | 6.2 | 30 | 500 | 4440.81±141.02e |
| 7 | 6.2 | 32 | 600 | 2713.69±151.29ab |
| 8 | 6.2 | 30 | 500 | 4175.04±130.20e |
| 9 | 6.2 | 28 | 600 | 4271.13±152.86e |
| 10 | 6.2 | 30 | 500 | 3451.09±157.22d |
| 11 | 6.6 | 30 | 400 | 2713.88±123.38ab |
| 12 | 5.8 | 30 | 400 | 3057.63±154.04abcd |
| 13 | 6.2 | 30 | 500 | 2840.01±136.11abc |
| 14 | 6.2 | 32 | 400 | 3040.75±140.81abcd |
| 15 | 6.6 | 30 | 600 | 4213.62±150.51e |
| 16 | 6.2 | 28 | 400 | 2691.53±133.36a |
| 17 | 6.2 | 30 | 500 | 3167.20±128.79bcd |

and the test variable. The second-order polynomial equation is as follows:

$$(Y) = 4241.40 + 125.81 \times A - 42.82 \times B + 106.60 \times C - 116.28 \times AB + 237.26 \times AC \\ + 138.71 \times BC - 634.68 \times A\hat{}2 - 627.73 \times B\hat{}2 - 592.25 \times C\hat{}2$$

**Variance analysis.** As shown in Table 7, the analysis of variance, adequacy, and goodness-of-fit of the regression model was performed with Design-Expert software (Version 8.0.6, Stat-

**Table 7. Variance analysis of the quadratic model by the response surface test of fermentation parameters.**

| Item | Quadratic sum | Degree of freedom | Mean square deviation | F-value | P-value |
|---|---|---|---|---|---|
| Model | 598905.40 | 9 | 66545.30 | 487.02 | < 0.0001 |
| A-pH | 12667.21 | 1 | 12667.21 | 92.70 | 0.0187 |
| B-temperature | 1463.23 | 1 | 1463.25 | 10.78 | 0.3352 |
| C-blades speed | 9089.48 | 1 | 9089.40 | 66.51 | 0.0365 |
| AB | 5411.85 | 1 | 5411.82 | 39.62 | 0.0869 |
| AC | 22527.73 | 1 | 22527.08 | 164.90 | 0.0048 |
| BC | 7696.98 | 1 | 7696.93 | 56.37 | 0.0494 |
| A2 | 169639.90 | 1 | 169639.93 | 1241.63 | < 0.0001 |
| B2 | 165919.0 | 1 | 165919.10 | 1214.42 | < 0.0001 |
| C2 | 147708.46 | 1 | 147708.41 | 1081.10 | < 0.0001 |
| Residual | 9564.11 | 7 | 1366.34 | | |
| Lack of fit | 3812.20 | 3 | 1270.71 | 0.88 | 0.5213 |
| Net error | 5751.93 | 4 | 1438.32 | | |
| Total error | 608469.66 | 16 | | | |
| $R^2$ | 0.9661 | | 0.62 | | |
| $R^2_{adj}$ | 0.9548 | | 0.71 | | |

Note: p<0.05, significant difference; p<0.01,very significant difference.

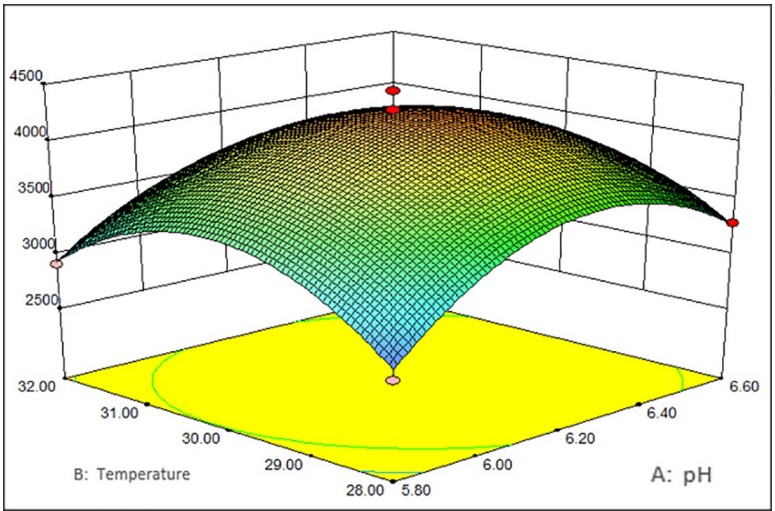

**Fig 1. Interaction of temperature and pH vs. the yield of MRYP.**

Ease, Inc., Minneapolis, NE, USA). The model was highly statistically significant because of a low P-value (P< 0.0001) and a high F-value (48.70). The high $R_2$ value of 0.9548 and the high $R_{2adj}$ value of 0.9661 indicate that the model was applicable and accurate. It also reveals that the interactions of pH and temperature, pH and blades speed, temperature and blades speed were insignificant (p > 0.05).

**Response surface analysis.** As shown in Figs 1–3, the corresponding three-dimensional response surfaces of MRYP yield and the different independent parameters were plotted, which elucidated the interaction of variables with the uttermost response. The middle values of the three parameters (pH value of 6.2 (A), blades speed of 500 r/min (B), temperature of 30°C (C)) were used to create the graphs. The sequence of factors that affected the MRYP yield was the blades speed>pH>temperature. While one factor was maintained at a fixed value, the effects of the other two factors on the polysaccharide yield were shown.

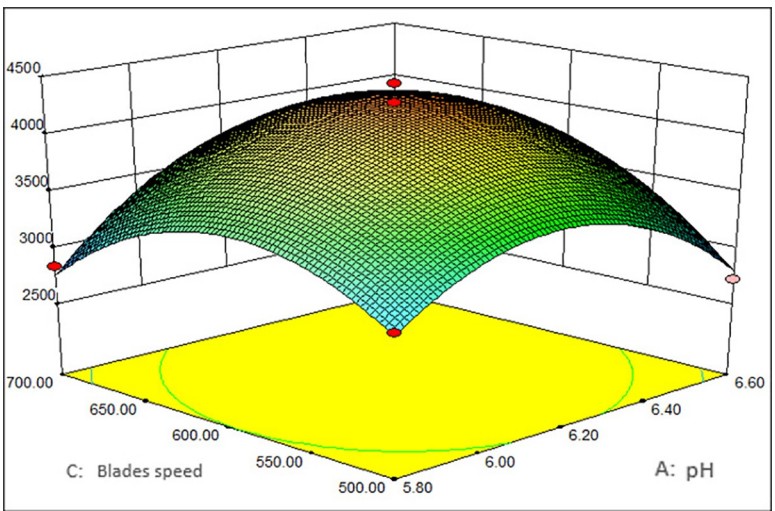

**Fig 2. Interaction of blades speed and pH vs. the yield of MRYP.**

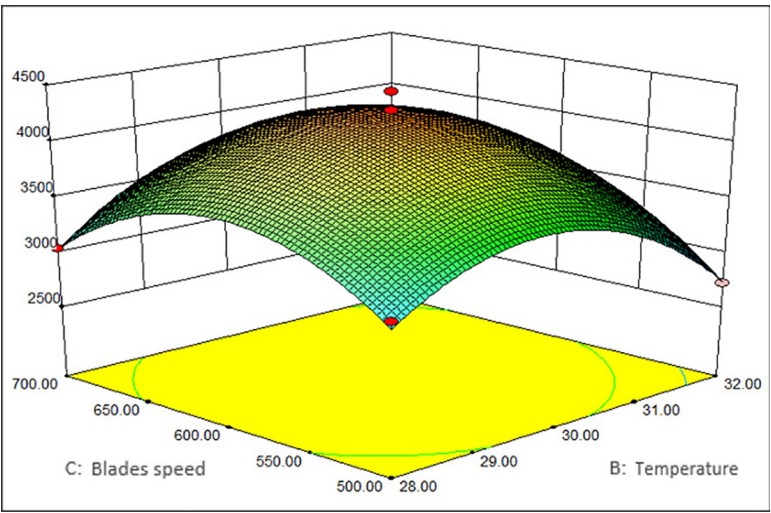

**Fig 3. Interaction of temperature and blades speed vs. the yield of MRYP.**

Fig 1 illustrates the polysaccharide yield in response to pH and temperature with a fixed blades speed of 500 r/min. The MRYP yield increased with an increase in pH and temperature until it reached its peak. After attaining the peak value, a further increase in pH and temperature led to a decreased polysaccharide yield. The effect of pH and blades speed on the polysaccharide yield is illustrated in Fig 2. When the temperature was 30°C, an increase in both pH and blades speed led to an increase in the polysaccharide yield at the beginning. After reaching the peak yield, the response surface decreased, and similar trends were observed in the effect of blades speed and temperature (Fig 3). Besides, the response surface analysis showed that pH, blades speed, and temperature showed significant interactive effects. According to the model, the maximum yield of MRYP was 4876.80 mg/L when the values of corresponding variables were pH = 6.72, temperature = 30.18°C, and blades speed = 461.36 r/min.

**Validation test.** The optimal fermentation conditions had been obtained in the prediction model, but we had to validate precisely how well the parameters work in practice. In view of the operating convenience, the optimal fermentation parameters were set to pH = 6.8, temperature 30°C, and blades speed 460 r/min.

The predicted MRYP yield was verified by validation tests after three replications. As shown in Fig 4, the average MRYP yield was 4323.90 mg/L, with a relative error of 11.34%. Thus, the average MRYP yield was close to the predicted yield. This result indicated that the model can be used to guide actual practice and production. The optimal yield of MRYP was 1.31 times higher than the yield before optimization, which was 3300.00 mg/L.

## Determination of the molecular weight

As shown in Fig 5, the pure polysaccharide was fractionated by using a DEAE-52 column and Sephadex G-100 column. The two main components of MRYP fractions were eluted, and each component produced a single and symmetrical sharp peak. The two main components were named MYH-1, MYH-2. Then the components were vacuum lyophilized and weighed precisely. The proportions of MYH-1 and MYH-2 in pure MRYP were 53.60% and 28.75%, respectively.

The molecular weight of MYH-1 and MYH-2 was determined by high-efficiency gel permeation chromatography (HPGPC). As shown in Fig 6, two single and symmetrical peaks were

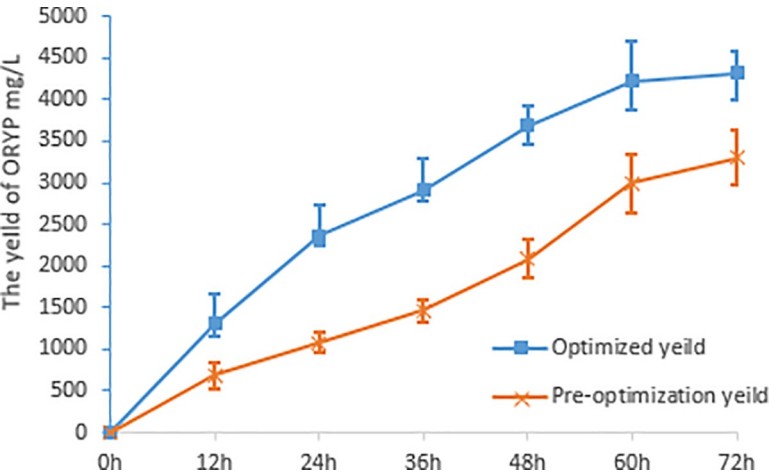

**Fig 4. The MRYP yields of optimized conditions and pre-optimization.**

observed in the HPGPC chromatogram. This result indicated that the components MYH-1 and MYH-2 were pure enough for the next analysis. The retention time of MYH-1 and MYH-2 was 14.432 min and 20.587 min, respectively. Fig 7 shows the standard curve of molecular weight. The retention time of MYH-1 and MYH-2 was included in the regression equation, and the molecular weight of MYH-1 and MYH-2 was found to be 246.92 kDa and 21.88 kDa.

## Analysis of the monosaccharides composition

The retention time and the standard curve of MYH-1 and MYH-2 were obtained by HPLC. Fig 8 shows the standard curve of glucose and mannose. As shown in Figs 9 and 10, the prominent peak retention time of glucose and mannose standard samples was 26.384 min and 30.217 min, respectively. Figs 11 and 12 show the chromatographic peak areas of glucose and mannose in MYH-1 and MYH-2 components, respectively. According to the regression equation, the concentration of glucose and mannose was determined. The glucose and mannose content in MYH-1 and MYH-2 was 938.72 μg/mL, 769.15 μg/mL and 1341.97 μg/mL,

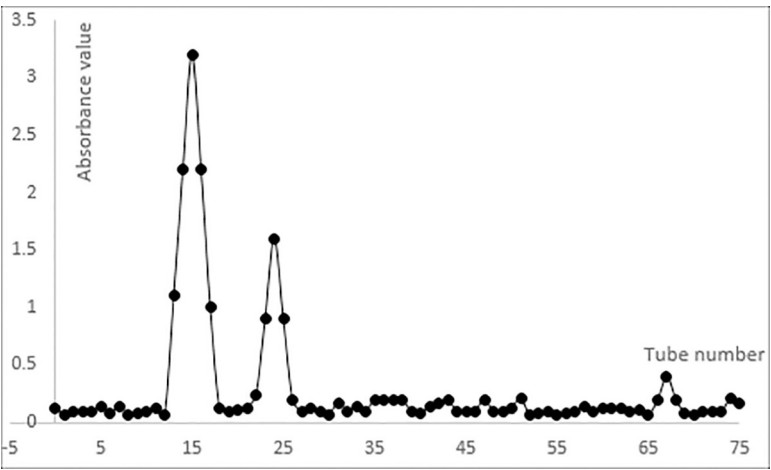

**Fig 5. Linear elution of pure polysaccharides.**

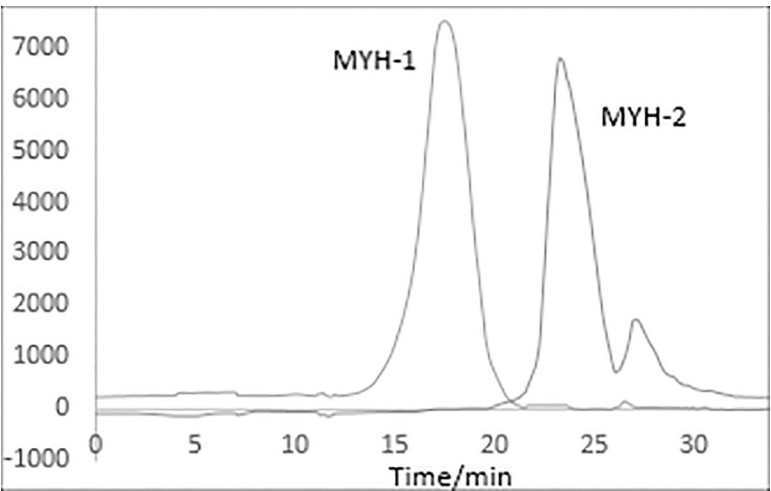

**Fig 6. HPGPC chromatograms of MYH-1 and MYH-2.**

143.46 μg/mL, respectively. The proportion of glucose and mannose in MYH-1 and MYH-2 was 46.94%, 38.46%, and 67.10%, 7.17%, respectively.

### *In vitro* tests

**Antioxidant activity.** Fig 13 shows the antioxidant ability of MRYP towards DPPH, OH, and $\cdot O_2^-$. It shows that there was an obvious dose-response relationship with the antioxidant ability of MRYP. The antioxidant ability increased gently until the concentration of MRYP approached 300 μg/mL. Then the antioxidant ability sharply increased and reached a peak while the concentration of MRYP was 1000 μg/mL. Besides, the antioxidant ability of MRYP towards DPPH· displayed higher values than others, and the clearance rate of MRYP to ·DPPH, OH, and $\cdot O_2^-$ was 32.26%, 24.34%, and 22.09%, respectively.

**Antibacterial activity.** The antibacterial activity of MRYP involved resistance to three kinds of pathogenic bacteria, namely, *Escherichia coli*, *Salmonella*, and *Staphylococcus aureus*, as shown in Table 8.

The growth of the three pathogenic bacteria was inhibited when the concentration of MRYP mixtures was 480 μg/mg. The antibacterial activity of MRYP on *Salmonella* was more

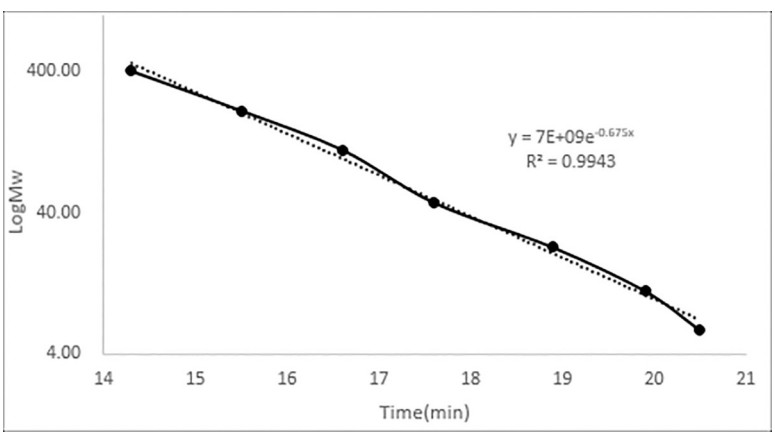

**Fig 7. Standard curve of polysaccharide of the standard molecular weight.**

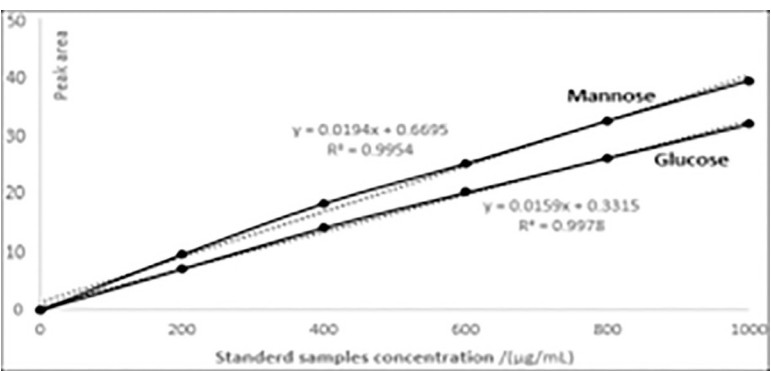

**Fig 8. Standard curve of glucose and mannose.**

effective than that on *Escherichia coli* and *Staphylococcus aureus*, with the inhibition zone diameters being 14.54±0.85 mm, 10.40±2.25 mm, and 9.53±0.11 mm. It did not show a bacteriostatic effect on any pathogenic bacteria when the mixture concentration was below 480 μg/mg. In this test, amoxicillin (200 μg/mL) showed a high antibacterial activity on the three pathogenic bacteria, and the inhibition zones diameters of *Escherichia coli*, *Salmonella*, and *Staphylococcus aureus* were 24.74±2.36 mm, 24.67±2.43 mm, and 24.43±2.44 mm, respectively. This result indicated that MRYP exhibits antibacterial activity on common pathogens, but the bacteriostatic effect of MRYP is weaker than that of antibiotics.

### *In vivo* tests

The objective of these experiments was to investigate the effects of MRYP on the growth, serum antioxidant enzyme activity, and feces bacteria in growing lambs.

**Effect of MRYP on lamb growth.**   Every experimental lamb was taken good care of by technicians, and no mortality was observed in the whole period. As shown in Table 9, 32-days-old lambs were weighed individually, and there was no significant difference in the weight of 32-days old lambs included in this study. At the end of the experiment, 60-days-old lambs were weighed individually again, and the weight of lambs in the antibiotic group (14.21±0.33 kg) was significantly greater than those of other groups. However, the weight of MRYP 0.3% group (13.43±0.20 kg) and MRYP 0.5% group (13.37±0.13 kg) was significantly greater than that of the control group (11.44±1.10 kg). This indicates that lamb's weight increased when MRYP was used as a feed additive. Besides, the standard deviation in the weight of MRYP 0.3% group and MRYP 0.5% group was lower than those of other groups, which implied that a

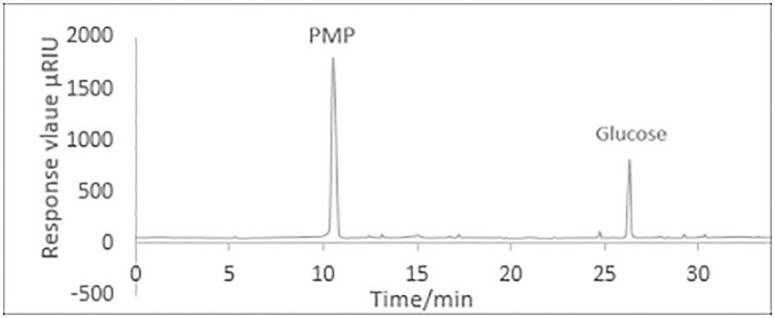

**Fig 9. HPLC chromatogram of glucose.**

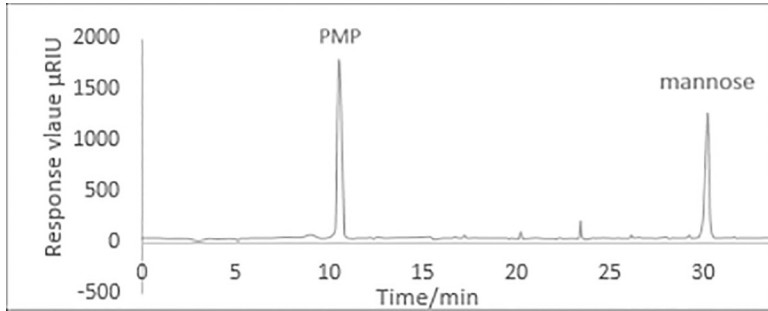

**Fig 10. HPLC chromatogram of mannose.**

certain amount of MRYP improved uniformity in the weight of lambs. In addition, the F/G of MRYP 0.3% was lower than the others, which indicates that MRYP might improve the production efficiency of lamb breeding.

**Effect of MRYP on the antioxidant activity of lamb serum.** Table 10 shows the antioxidant activity of 60-days-old lamb serum. Four antioxidant enzymes did not show a significant difference in all the groups. But the antioxidant activity of GSH-Px, T-SOD, and T-AOC in MRYP 0.3% group had higher values than in other groups, which indicates that MRYP increased the antioxidant enzyme activity to a certain extent. Besides, the content of MAD in MRYP 0.3% was lower than in the control group, which indicates that MRYP can prevent damage to cell membranes. The result agreed with the conclusion of antioxidant activity in vitro test.

**Effect of MRYP on bacteria quantity in lamb feces.** Table 11 shows the number of bacteria in lamb faeces. It was obvious that the number of *Bifidobacterium* and *Lactobacillus* in MRYP 0.3% group was significantly higher than in other groups. This indicates that MRYP increased the number of probiotics in lamb's intestines. The number of *Escherichia coli* and *Salmonella* was significantly lower in MRYP 0.3% group than in the control group, which indicates that MRYP inhibited the growth of pathogens in lamb's intestines. This result was approximately the same as in the *in vitro* experiment of antibacterial activity.

## Discussion

In this paper, we proved that Medium 4 was the optimal medium for fermentation, but the formulation was different from common media for yeast, such as PDA, YPD, and malt extract medium. This indicates that the nutritional requirement was different in different kinds of

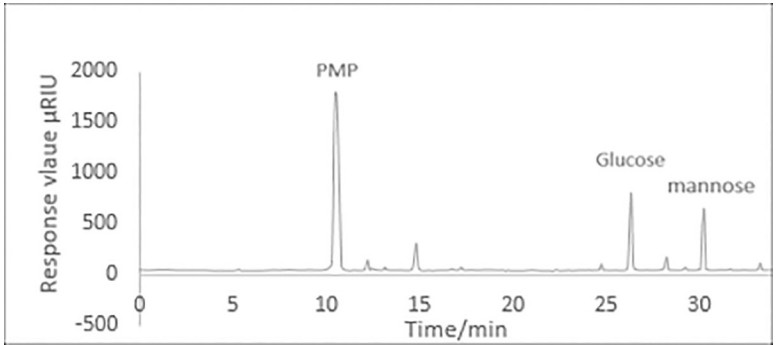

**Fig 11. HPLC chromatogram of the MYH-1 hydrolysate.**

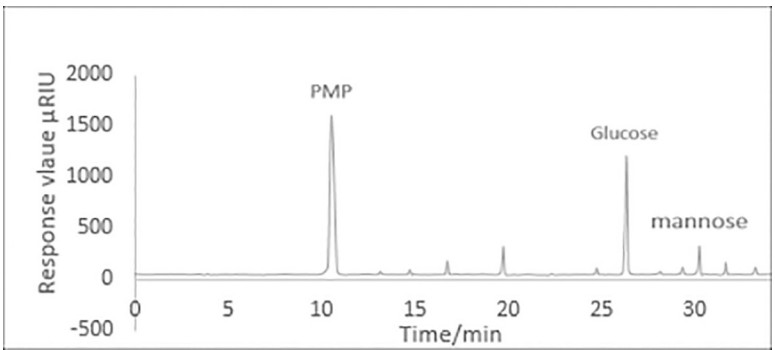

**Fig 12. HPLC chromatogram of the MYH-2 hydrolysate.**

yeast strains and fermentation products. Furthermore, the optimal fermentation parameters for MRYP were pH = 6.72, temperature = 30.18˚C, blades speed = 461.36 r/min, and the optimized yield reached 432.39 mg/100 mL, which was 1.31 times the original yield. There are some reports concerning the optimal fermentation parameters for yeast polysaccharides. It has been reported that a 0.20 g $L^{-1}$ $h^{-1}$ polysaccharide that was composed of 18.57% (w/w) kefiran was obtained by using optimal fermentation parameters (whey supplemented with 15% (w/v) glucose and fermented at 30˚C for 10 h) [52]. The fermentation conditions of red yeast *Rhodotorula* were optimized as follows: 0.5% yeast powder, 2% molasses, 50 mL of a medium in a 250 mL flask, and temperature 20˚C, initial pH = 3.0, cultivation for 96 h [53]. The optimal fermentation conditions of Monascus sp. were determined as follows: yeast extract powder 45 g/L, sucrose 45 g/L, $MgSO_4 \cdot 7H_2O$ 0.85g/L, $KH_2PO_4 \cdot 3H_2O$ 35 g/L, and inoculation volume 75%, seed culture age 30 h, initial pH 5.75, and fermentation for 84 h. Under the optimal fermentation conditions of red yeast *Rhodotorula*, the highest yield of polysaccharide was 999.8 mg/L and improved by 46.1% compared with the original yield of 684.2 mg/L [54]. In this experiment, the sequence of factors (blades speed>pH>temperature) was different from that in the pilot experiment (temperature>blades speed>pH). This is because the optimal pH scope was minimized in the primary screening test, which led to a pH-insensitive yield of MRYP. Furthermore, the oxygen content in the fermentation solution was different in a fermenter with rotating blades and a shaking flask in the pilot experiment. Because the optimized yield was

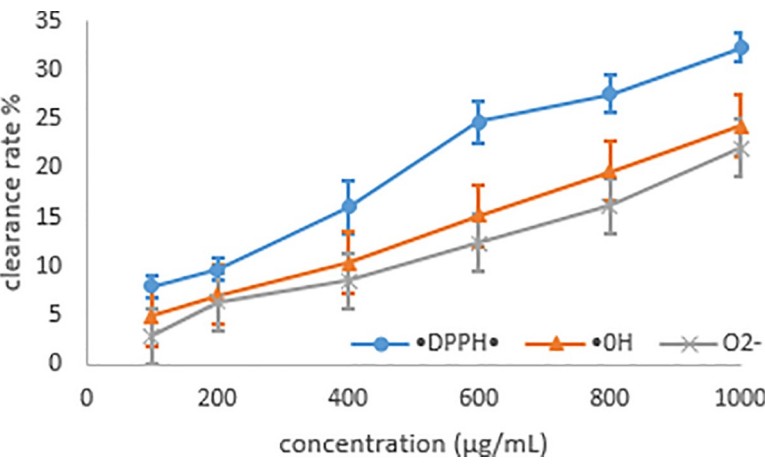

**Fig 13. Scavenging capacities of MRYP for DPPH, OH, and ·O2−.**

**Table 8. The average diameter of the inhibition zone of 3 pathogenic bacteria (mm).**

| Test strain | MRYP (μg/mg) | | | | | Amoxicillin | Sterile water |
|---|---|---|---|---|---|---|---|
| | 30 | 60 | 120 | 240 | 480 | | |
| Escherichia coli | - | - | - | - | 10.40±2.25 | 24.74±2.36 | - |
| Salmonella | - | - | - | - | 14.54±0.85 | 24.67±2.43 | - |
| Staphylococcus aureus | - | - | - | - | 9.53±0.11 | 24.43±2.44 | - |

below expectations, the screening of high-MRYP-yield yeast strain and genetically engineered yeast was included in our next project. The optimal temperature in this experiment was higher than that reported in previous research studies [55]. This is because the experimental marine red yeast Rhodosporidium paludigenum strain was collected from Yarkand River in Kashi region, which is a parched area that supports the thermophilic feature of Rhodosporidium paludigenum strain.

As is well known, the biological properties of polysaccharides depend on the molecular weight, chemical composition, and configuration. The molecular weight as the most fundamental indicator should be investigated to indicate the bioactivity of polysaccharides. The higher molecular weight of acidic polysaccharides, the higher activity in the biological complement system the polysaccharides have. It has been reported previously that only polysaccharides with a molecular weight of 50.0 kDa to 200.0 kDa have biological activities, normally [9]. Besides, there is a report suggesting that the polysaccharides with the molecular weight in the range from 4.0 kDa to 100.0 kDa expressed a better antioxidant activity. In addition, there are some differences in the structural characterization and chemical composition of polysaccharides, which may be related to the yeast species and fermentation conditions. Some studies have proved that glucose and mannose are the major bioactive components in polysaccharides, which widely exist in yeast, bacteria, algae, and plants and have been well known for their bioactivities including wound healing, antibacterial, enhancing immunity, and antioxidant activities [56]. In this study, the molecular weights of MYH-1 and MYH-2 were 246.92 kDa and 21.88 kDa, respectively, and they accounted for 53.60% and 28.75% of total polysaccharide. Besides, the proportion of glucose and mannose accounted for 46.94%, 38.46%, and 67.10%, 7.17% in MYH-1 and MYH-2, respectively. Therefore, the MRYP probably expresses some biological activities. It has been reported that the GIP-II was the main fraction in polysaccharides from Gleoestereum incarnatum and was composed of mannose, glucose, galactose, and xylose [57]. In another study, the major purified polysaccharides in a mulberry leaf polysaccharide were MLP-1 and MLP-2, which were composed of glucose, mannose, galactose, rhamnose, and arabinose. The molecular weights of MLP-1 and MLP-2 were 222.0 kDa and 93.1 kDa, respectively [58]. The polysaccharides from chestnut (Castanea mollissima Bl.) kernel were extracted by using successive ultrasound-assisted-extraction, and the major fractions UEP1-1 and UEP2-1 were obtained. The molecular weights of UEP1-1 and UEP2-1 were 74.5

**Table 9. Effect of MRYP on body weight.**

| Items | MRYP level (%) | | | | |
|---|---|---|---|---|---|
| | Control group | Antibiotic group | MRYP 0.1% | MRYP 0.3% | MRYP 0.5% |
| 32-days old weight | 7.67±1.38 | 8.89±1.22 | 8.31±1.42 | 8.36±1.26 | 8.23±1.11 |
| 60-days old weight | 11.44±1.10a | 14.01±0.73c | 12.86±0.51a | 13.43±0.20b | 13.37±0.13b |
| Average daily gain(g) | 134.64±16.19a | 182.86±15.88c | 162.5±13.42b | 181.07±11.41c | 183.57±12.52c |
| Average daily feed intake (g) | 373.50±23.51a | 405.27±19.54b | 396.21±16.46b | 398.79±11.15b | 429.53±12.40b |
| F/G | 2.77 | 2.21 | 2.44 | 2.20 | 2.34 |

**Table 10. Effect of MRYP on the serum antioxidant enzymes activity of lambs.**

| Items | MRYP levels (%) | | |
|---|---|---|---|
| | Control group | Antibiotic group | MRYP 0.3% |
| GSH-Px (μmol/L) | 15.45±1.16 | 14.83±0.31 | 15.79±0.95 |
| T-SOD(U/mL) | 1.34±0.43 | 1.38±0.45 | 1.42±0.56 |
| MAD (nmol/mL) | 2.57±0.11 | 2.46±0.43 | 2.33±0.65 |
| T-AOC(U/mL) | 2.47±0.31 | 2.50±0.63 | 2.61±0.42 |

kDa and 59.9 kDa, respectively, and the monosaccharide composition of UEP1-1 and UEP2-1 was glucose, mannose, rhamnose, galactose, and arabinose.

The studies have proven that the antioxidant and antibacterial activities mainly depend on physicochemical features including the molecular weight and monosaccharide composition [59]. The ·DPPH,·OH, and·$O_2^-$ scavenging percentages were used to indicate the antioxidant ability of polysaccharides *in vitro*. The free radical activities were increased by reduction of antioxidant enzymatic activities, therefore, antioxidant enzymatic activities could indicate the radical activities and oxidative stress *in vivo*. *In vitro*, the antioxidant activity of MRYP was weak and far below that of vitamin C. The same result was found *in vivo*, MRYP did not result in a significant difference in the antioxidant activity of GSH-Px, T-SOD, and T-AOC in the lamb serum. This implies that the antioxidant activity is not the main biological function of intestinal tract, and MRYP should not be used as an antioxidant in lambs.

However, *in vitro*, MRYP strongly inhibited common pathogens, especially Salmonella. This result was approximately the same as the result of effect of MRYP on bacteria quantity in lamb feces according to which the supplementation of 0.3% MRYP significantly increased the number of *Bifidobacterium* and *Lactobacillus* but decreased the number of *Escherichia coli* and *Salmonella*. This is a meaningful result to encourage the microflora balance of intestines and to promote the healthy function of the intestinal system in non-reactive breeding grounds. It has been reported that association of mannose and NAC therapy resulted similarly in antibiotic therapy in preventing UTIs in patients submitted to urodynamic examination [60]. D-mannose, N-acetylcysteine, and Morinda citrifolia fruit extract showed a greater efficacy in reducing urinary discomfort and urinary tract infections with respect to antibiotic use only in breast cancer survivor women affected by genitourinary discomfort [61]. It has been also reported that β-glucan has lots of biological activities, such as antimicrobial, antioxidative, cardioprotective, immunomodulatory, and hepatoprotective activities [62]. The reports showed that β-glucan extracted from yeast cell walls had obvious hydroxyl radical, superoxide anion scavenging activities and anti-lipid peroxidation effect [63].

It has been proved that oxidative stress is harmful for body, and over-produced reactive oxygen species (ROS) cannot be scavenged by the antioxidant system in our body, which can result in enteritis, diarrhea, and cardiovascular diseases. On the other hand, antibiotic resistance is becoming a global problem, and about 70% of known pathogenic bacteria are resistant

**Table 11. Effect of MRYP on five kinds of bacteria in lamb feces (log CFU/g).**

| Items | MRYP levels (%) | | |
|---|---|---|---|
| | Control group | Antibiotic group | MRYP 0.3% |
| Bifidobacterium | 6.26±0.15b | 5.55±0.27a | 6.84±0.13c |
| Lactobacillus | 7.34±0.28a | 6.54±0.20a | 7.78±0.29b |
| Escherichia coli | 6.73±0.08b | 5.62±0.25a | 6.03±0.14a |
| Salmonella | 4.77±0.35c | 3.34±0.41a | 3.93±0.56b |
| Staphylococcus aureus | 4.62±0.80 | 4.09±0.97 | 4.51±0.91 |

to at least one antibiotic [64]. As is well known, *Bifidobacterium* and *Lactobacillus* are probiotics in intestines, but *Escherichia coli* and *Salmonella* can cause diarrhea and enteritis. *In vivo*, the lambs of the MRYP-trial group were fed a certain MRYP amount instead of antibiotics. All the lambs were physically fit, and none of them died in the whole experiment, which proves that MRYP is safe enough to be used in farming production. Besides, a lower F/G in lambs of MRYP-trial group than in the control group indicates that the production efficiency in sheep breeding could be improved partially. However, further research studies must be conducted to verify the conclusions of this experiment. In addition, there was not a significant difference in antioxidant activities of lamb serum.

The process of extraction and purification played an important role in the yield and biological function of MRYP. Based on the data of this experimental study, there is a close relationship between extraction conditions and the yield and biological functions of MRYP. These factors must be explored for providing a theoretical foundation for possible application in industrial production.

## Conclusion

In this study, the optimal fermentation cultivation medium and parameters of the production of MRYP were as follows: Medium 4, pH = 6.72, temperature 30.18˚C, and blades speed 461.36 r/min. The optimized yield of MRYP was 4323.90 mg/L. The main components of MRYP were MYH-1 and MYH-2. The molecular weight of MYH-1 and MYH-2 was 246.92 kDa and 21.88 kDa. The proportion of glucose and mannose in MYH-1 and MYH-2 was 46.94%, 38.46%, and 67.10%, 7.17%, respectively.

*In vivo*, the MRYP increased the growth of lambs. The antioxidant activity of MRYP was weak, but the antibacterial activity of MRYP strongly inhibited the three common pathogens both *in vitro* and *vivo*.

This work provides a theoretical foundation to a wide range of possible application of MRYP in the industrial production, design, and operation.

Subsequent work should investigate the effect of MRYP on the intestinal flora in lambs by genomics. The results will provide a theoretical basis for the production and application of MRYP to enhance productivity in animal industries.

## Author Contributions

**Data curation:** Mengjian Liu, Junli Niu.

**Formal analysis:** Mengjian Liu, Jun Yao.

**Funding acquisition:** Mengjian Liu.

**Investigation:** Mengjian Liu.

**Methodology:** Mengjian Liu, WenJu Zhang.

**Project administration:** WenJu Zhang.

**Supervision:** WenJu Zhang.

**Writing – original draft:** Mengjian Liu.

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
