## [Decision Letter · Decision Letter 0]

8 Jun 2020

PONE-D-20-05171

Fermentation Optimization and Purification of Oceanic Red Yeast Polysaccharide and Study on its Characterization and Biological Properties

PLOS ONE

Dear Dr. Liu,

Thank you for submitting your manuscript to PLOS ONE. After careful consideration, we feel that it has merit but does not fully meet PLOS ONE’s publication criteria as it currently stands. Therefore, we invite you to submit a revised version of the manuscript that addresses the points raised during the review process.

We look forward to receiving your revised manuscript.

Kind regards,

Chenyu Du, Ph.D.

Academic Editor

PLOS ONE

2. Thank you for including the following ethics statement in your cover letter:

'All experimental lambs in this research have been prospectively approved and granted a

formal waiver of ethics approval by the Animal Welfare Committee of Shihezi University

(Xinjiang, China) with the ethical code: A2019-156-01.

In the antioxidant enzyme activity test, three blood samples (10 mL each) were collected

randomly from the marginal ear vessels of 10 lambs in each group, and without anesthesia,

euthanasia, or any kind of animal sacrifice is part of the study.'

Please also include this information in the ethics statement in the Methods section of your manuscript.

4. Please upload a copy of Supporting Information Figure 1-13 which you refer to in your text on page 67.

Reviewers' comments:

Reviewer's Responses to Questions

**Comments to the Author**

1. Is the manuscript technically sound, and do the data support the conclusions?

Reviewer #1: Yes

Reviewer #2: Yes

2. Has the statistical analysis been performed appropriately and rigorously? 

Reviewer #1: Yes

Reviewer #2: I Don't Know

3. Have the authors made all data underlying the findings in their manuscript fully available?

Reviewer #1: Yes

Reviewer #2: Yes

4. Is the manuscript presented in an intelligible fashion and written in standard English?

Reviewer #1: Yes

Reviewer #2: Yes

5. Review Comments to the Author

Reviewer #1: In this manuscript, the author has explored biological properties including antioxidant and antibacterial activities of ORYP in vitro and vivo. The work deserves recognition. But there are some question needed to be addressed.

1. In part of “Optimization of fermentation medium”, what is method of 1.3.7? The author needs to verify this information.

2. Some mistakes should be corrected as follows:

In part of “Optimization of fermentation medium”, “asfollows” should be “as follows”;

In part of “Isolation of ORYP”, “werecollected” should be “were collected”;

In part of “Effects of ORYP on lambs in vitro”, “in vitro” should be “in vivo”;

Besides, “p ＜ 0.05” should be “p ＜ 0.05”. Many details may need some modifications.

3. In part of “Antibacterial activity”, the antibacterial activity is suggested to be shown by minimum inhibitory concentration and inhibitory concentration, which are more persuasive of its antibacterial activity.

4. In part of “Discussion”, this viewpoint “It has been reported previously that only polysaccharides with a molecular weight of 10.0kDa to 200.0 kDa have biological activity, normally.” is supposed to be supported by more relative literatures.

5. In part of “Optimization by response surface methodology”, the author has put too much attention on analytical process, which can be shortened.

6. The author has studied antioxidant activities of ORYP in vitro and in vivo, but these indicators are lack of connection. I suggest the author should give more discussion about the relationship among these indicators.

7. The author has studied the effect of ORYP on bacteria in vitro and in vivo, the results of which also need more deep discussion on these indicators.

Reviewer #2: This is an interesting manuscript contains some significant data and scientific effort. I recommend this paper for publication in the PLOSONE journal after careful consideration of the following comments: (Major Revision)

1- (Optimal) I suggest changing the title to: Production, Purification, Characterization and Biological Properties of Oceanic Red Yeast Polysaccharide OR Characterization and Biological Properties of Oceanic Red Yeast Polysaccharide

2- It is extremely poor practice when the author provides a manuscript with pages and lines not numbered, the text not justified and the space between lines are not 1.5 at least.

3- What is ‘Oceanic Red Yeast’? Do you mean marine the red yeast Rhodosporidium paludigenum?. If so, you must change, if not then you need to explain what is Oceanic Red Yeast’ in the Abstract and introduce it well in the introduction.

4- Reporting yield in mg/100mL is very confusing. Report as g/L or mg/L

5- How did you obtain this yeast from the River? Have you isolated it? If so, how? And how have you identified it?. If you obtain from a culture collection, provide details. If you bought it as a food grade commercial product, give details.

6- From the name that you suggested ‘Oceanic Red Yeast’, this yeast must be a marine strain. Marine yeasts have not been yet intensively investigated and therefore, you should highlighted this in introduction. Recent research suggests that Marine microorganisms have superior characteristics compared to their terrestrial counterparts. Elaborate on that (in the introduction and discussion sections) using appropriate references including the following (1-6)

7- You should insert page number and line number so reviewers can refer to them easily.

8- What is ‘the method of 1.3.7’ under ‘Optimization of fermentation medium’

9- Revise table 1. It seems you swapped the last two columns

10- There are some errors but it is hard to state here, as I cannot refer the author to page number and line number, they are not provided!

11- Under ‘Effect of ORYP on antioxidant enzyme activity of lamb serum’ the author state: ‘…The result agreed with the conclusion of chapter 2.8.1. ....’. Well, I cannot see these chapters! There are other places were author refer to chapters do not exist.

12- Many linguistic and technical errors in the ‘Discussion’ section, this need to be revised carefully. The English of this section in particular needs improving.

13- The conclusion section lacks the conclusion! The significance of this work and the future prospective should be provided.

14- Error bars should be included in tow of the Figures, Fig. 4 and Fig. 13.

6. PLOS authors have the option to publish the peer review history of their article (what does this mean?). If published, this will include your full peer review and any attached files.

Reviewer #1: No

Reviewer #2: No

---

## [Author Response · Author response to Decision Letter 0]

9 Oct 2020

Thank you for the suggestion. All suggestions, which specific reviewer and editor advised have been changed in my manuscropt.

The responds and changes have been shown 

specifically in files "Response to Reviewers #1" and "Response to Reviewers #2", which have been uploaded in this system.

---

## [Decision Letter · Decision Letter 1]

6 Nov 2020

PONE-D-20-05171R1

Production, Purification, Characterization and Biological Properties of Marine Red Yeast Rhodosporidium Mucilaginosa Polysaccharide

PLOS ONE

Dear Dr. Liu,

Thank you for submitting your revised manuscript to PLOS ONE. However, a response letter with detailed replied to reviewers' comments is required before it could be further considered. And, please add line number and page number as well in the revised manuscript. 

We look forward to receiving your revised manuscript.

Kind regards,

Chenyu Du, Ph.D.

Academic Editor

PLOS ONE

Reviewers' comments:

Reviewer's Responses to Questions

**Comments to the Author**

1. If the authors have adequately addressed your comments raised in a previous round of review and you feel that this manuscript is now acceptable for publication, you may indicate that here to bypass the “Comments to the Author” section, enter your conflict of interest statement in the “Confidential to Editor” section, and submit your "Accept" recommendation.

Reviewer #2: (No Response)

2. Is the manuscript technically sound, and do the data support the conclusions?

Reviewer #2: (No Response)

3. Has the statistical analysis been performed appropriately and rigorously? 

Reviewer #2: (No Response)

4. Have the authors made all data underlying the findings in their manuscript fully available?

Reviewer #2: (No Response)

5. Is the manuscript presented in an intelligible fashion and written in standard English?

Reviewer #2: (No Response)

6. Review Comments to the Author

Reviewer #2: It is hard to assess the revision. The author has not provided letter of response for point-by-point.

In order to assess the revised manuscript the author should do the following:

1- Submit revised manuscript that contains line number (continues), page number, changes and the added text are highlighted or typed in a different colour.

2- Submit a response letter to explain point-by-point the details of the revisions in the manuscript. Page and line number where changes were made or text added in the revised manuscript must be provided with each point. A rebuttal for points that impossible to be addressed.

The previous comments are included again.

This is an interesting manuscript contains some significant data and scientific effort. I recommend this paper for publication in the PLOSONE journal after careful consideration of the following comments: (Major Revision)

1- (Optimal) I suggest changing the title to: Production, Purification, Characterization and Biological Properties of Oceanic Red Yeast Polysaccharide OR Characterization and Biological Properties of Oceanic Red Yeast Polysaccharide

2- It is extremely poor practice when the author provides a manuscript with pages and lines not numbered, the text not justified and the space between lines are not 1.5 at least.

3- What is ‘Oceanic Red Yeast’? Do you mean marine the red yeast Rhodosporidium paludigenum?. If so, you must change, if not then you need to explain what is Oceanic Red Yeast’ in the Abstract and introduce it well in the introduction.

4- Reporting yield in mg/100mL is very confusing. Report as g/L or mg/L

5- How did you obtain this yeast from the River? Have you isolated it? If so, how? And how have you identified it?. If you obtain from a culture collection, provide details. If you bought it as a food grade commercial product, give details.

6- From the name that you suggested ‘Oceanic Red Yeast’, this yeast must be a marine strain. Marine yeasts have not been yet intensively investigated and therefore, you should highlighted this in introduction. Recent research suggests that Marine microorganisms have superior characteristics compared to their terrestrial counterparts. Elaborate on that (in the introduction and discussion sections) using appropriate references including the following:

References:

- Zaky, A.S., et al., The establishment of a marine focused biorefinery for bioethanol production using seawater and a novel marine yeast strain. Scientific Reports, 2018. 8(1): p. 12127.

- Zaky, A.S., et al., Marine yeast isolation and industrial application. FEMS Yeast Res, 2014. 14(6): p. 813-25.

- Zaky, A., et al., A New Isolation and Evaluation Method for Marine Derived Yeast spp with Potential Applications in Industrial Biotechnology. Journal of microbiology and biotechnology, 2016. 26(11): p. 1891-1907.

- Greetham, D., A.S. Zaky, and C. Du, Exploring the tolerance of marine yeast to inhibitory compounds for improving bioethanol production. Sustainable Energy & Fuels, 2019.

- A.S. Zaky* C. E. French, G.A. Tucker and C. Du (2020). Improving the productivity of bioethanol production using marine yeast and seawater-based media. Biomass & Bioenergy. 139, 2020,105615, https://doi.org/10.1016/j.biombioe.2020.105615.

- Fotedar, R., et al Fungal diversity of the hypersaline Inland Sea in Qatar https://doi.org/10.1515/bot-2018-0048

7- You should insert page number and line number so reviewers can refer to them easily.

8- What is ‘the method of 1.3.7’ under ‘Optimization of fermentation medium’

9- Revise table 1. It seems you swapped the last two columns

10- There are some errors but it is hard to state here, as I cannot refer the author to page number and line number, they are not provided!

11- Under ‘Effect of ORYP on antioxidant enzyme activity of lamb serum’ the author state: ‘…The result agreed with the conclusion of chapter 2.8.1. ....’. Well, I cannot see these chapters! There are other places were author refer to chapters do not exist.

12- Many linguistic and technical errors in the ‘Discussion’ section, this need to be revised carefully. The English of this section in particular needs improving.

13- The conclusion section lacks the conclusion! The significance of this work and the future prospective should be provided

14- Error bars should be included in tow of the Figures, well Figure number is not provided too, this again show the poor practice and carelessness

7. PLOS authors have the option to publish the peer review history of their article (what does this mean?). If published, this will include your full peer review and any attached files.

Reviewer #2: No

---

## [Author Response · Author response to Decision Letter 1]

27 Nov 2020

Dear editor and reviewer, all detials of changes in my manuscript have been done in my file of "Response to Reviewers", which has been submitted by submission system. I will sent you a mial adhered with the files "Response to Reviewers", "Revised Manuscript with Track Changes" and "Manuscript". Thank you for your time. With my best regards.

---

## [Decision Letter · Decision Letter 2]

29 Dec 2020

PONE-D-20-05171R2

Production, Purification, Characterization and Biological Properties of Marine Red Yeast Rhodosporidium Paludigenum Polysaccharide

PLOS ONE

Dear Dr. Liu,

Thank you for submitting your manuscript to PLOS ONE. After careful consideration, we feel that it has merit but does not fully meet PLOS ONE’s publication criteria as it currently stands. Therefore, we invite you to submit a revised version of the manuscript that addresses the points raised during the review process.

Editor's comment:

The language quality of the manuscript has still plenty of scope for improvement. It is, therefore, essential that before acceptance, the standard is suitable for publication.  I advise you to consult a native English speaker for assistance before submitting a revised version of your paper.

We look forward to receiving your revised manuscript.

Kind regards,

Chenyu Du, Ph.D.

Academic Editor

PLOS ONE

Additional Editor Comments (if provided):

The English should be improved. 

Reviewers' comments:

Reviewer's Responses to Questions

**Comments to the Author**

1. If the authors have adequately addressed your comments raised in a previous round of review and you feel that this manuscript is now acceptable for publication, you may indicate that here to bypass the “Comments to the Author” section, enter your conflict of interest statement in the “Confidential to Editor” section, and submit your "Accept" recommendation.

Reviewer #2: All comments have been addressed

2. Is the manuscript technically sound, and do the data support the conclusions?

Reviewer #2: Yes

3. Has the statistical analysis been performed appropriately and rigorously? 

Reviewer #2: (No Response)

4. Have the authors made all data underlying the findings in their manuscript fully available?

Reviewer #2: Yes

5. Is the manuscript presented in an intelligible fashion and written in standard English?

Reviewer #2: (No Response)

6. Review Comments to the Author

Reviewer #2: Again, the author failed miserably in providing a clear response to reviewer's comments. Submitted 3 confusing version of his manuscript and no relation between the page numbers provided and where changes made.

Going over the manuscript I can see that the author have made some improvement to the manuscript however, the author needs training on how to write a coherent scientific paper and needs to take it more seriously! The current manuscript need a revision for experienced colleague of the author to meet the standard of international journals.

There are still many technical shortfalls spotted such as:

1- long keyword 'Marine red yeast rhodosporidium paludigenum polysaccharide'

2- inconsistence use of slash divider 'g/L' (line529), and superscript ' L-1 h-1'(line522) and so many places

3- Table 2. 'polysaccharide' should be added to the title and a,b,c should be explained on the legend

4- English errors such as 'Over more' in line 518

5- The English, especially the added parts, is poor. It needs revision for the form and coherence.

7. PLOS authors have the option to publish the peer review history of their article (what does this mean?). If published, this will include your full peer review and any attached files.

Reviewer #2: No

---

## [Author Response · Author response to Decision Letter 2]

6 Jan 2021

1. The whole article has been improved by native English professer. 

2. The key word 'Marine red rhodosporidium paludigenum polysaccharide' has been changed to ‘Rhodosporidium paludigenum polysaccharide’.

3. The explanation has been added under the Table2.

4. The 'Over more' in Line 505 has been changed to Furthermore.

---

## [Editor Report · Decision Letter 3]

15 Jan 2021

Production, Purification, Characterization, and Biological Properties of Rhodosporidium Paludigenum Polysaccharide

PONE-D-20-05171R3

Dear Dr. Liu,

We’re pleased to inform you that your manuscript has been judged scientifically suitable for publication and will be formally accepted for publication once it meets all outstanding technical requirements.

Kind regards,

Chenyu Du, Ph.D.

Academic Editor

PLOS ONE

Additional Editor Comments (optional):

I've checked the revision and it is significantly improved from the original manuscript.

---

## [Editor Report · Acceptance letter]

20 Jan 2021

PONE-D-20-05171R3 

Production, Purification, Characterization, and Biological Properties of Rhodosporidium Paludigenum Polysaccharide 

Dear Dr. Liu:

I'm pleased to inform you that your manuscript has been deemed suitable for publication in PLOS ONE. Congratulations! Your manuscript is now with our production department. 

Kind regards, 

on behalf of

Dr. Chenyu Du 

Academic Editor

PLOS ONE